# Adapting HFMCA to Graph Data: Self-Supervised Learning for Generalizable fMRI Representations

**Jakub Frąc**
Vrije Universiteit Amsterdam

**Qiang Li**
TReNDS Center

**Guido Van Wingen**
Amsterdam University Medical Center

**Shujian Yu**
Vrije Universiteit Amsterdam

## Abstract

Functional magnetic resonance imaging (fMRI) analysis faces major challenges due to limited data and variability across studies. Existing self-supervised methods from computer vision often rely on positive–negative pairs, which are difficult to define for neuroimaging data. We adapt the Hierarchical Functional Maximal Correlation Algorithm (HFMCA) to graph-structured fMRI, providing a principled framework that measures statistical dependence and enables robust self-supervised pretraining. Across five neuroimaging datasets, our method yields competitive embeddings for multiple classification tasks and transfers effectively to unseen domains. Code and supplementary material: https://github.com/fr30/mri-eigenencoder

## 1  Introduction

Functional magnetic resonance imaging (fMRI) reveals brain dynamics, with resting-state functional connectivity serving as a key biomarker for neurological and psychiatric disorders [15, 7, 13]. Deep learning, however, struggles with limited data, heterogeneous preprocessing, and domain shifts.

Self-supervised learning (SSL) methods adapted from computer vision [17, 20, 3, 14] address these issues. Graph-based approaches, modeling functional connectivity matrices, offer interpretable, low-dimensional representations and preserve network topology through augmentations.

The **H**ierarchical **F**unctional **M**aximal **C**orrelation **A**lgorithm (HFMCA) [11] extends this by measuring statistical dependence across multiple feature hierarchies without contrastive pairs [4, 10, 21, 1]. Operating on graph-structured data, it captures richer dependencies and yields more generalizable neuroimaging representations.

**Contributions:** (1) We adapt HFMCA to graph-structured fMRI data, representing the first application and extension of this framework to brain connectivity networks. (2) We demonstrate that HFMCA-pretrained encoders produce competitive embeddings for neuroimaging classification tasks across diverse datasets. (3) We show effective transfer learning capabilities, particularly in scenarios where limited labelled data is available. 4) We evaluate neural scaling laws in the context of fMRI graph encoders, showing that naive pretraining data scaling may induce negative transfer.

## 2  Background and Methods

### 2.1  Functional Maximal Correlation Algorithm

The Functional Maximal Correlation Algorithm (FMCA) [11] measures statistical dependence between two random variables $X$ and $Y$ by maximizing the correlation between their nonlinear

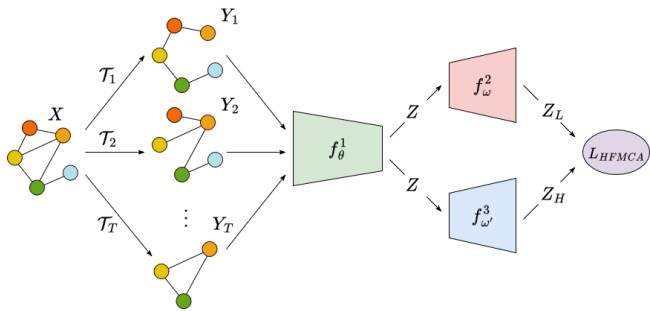

Figure 1: HFMCA learns graph representations by maximising dependence between low- and high-level features from multiple augmentations.

| Model | BSNIP | AOMIC (Sex) | HCP (Sex) |
|---|---|---|---|
| $\text{Baseline}_F$ | $47.7 \pm 0.9$ | $56.9 \pm 1.3$ | $63.0 \pm 2.4$ |
| $\text{VICReg}_F$ | $46.4 \pm 0.9$ | $\underline{60.0 \pm 1.5}$ | $65.7 \pm 1.5$ |
| $\text{BT}_F$ | $46.2 \pm 1.2$ | $\mathbf{60.9 \pm 0.8}$ | $\underline{66.0 \pm 1.5}$ |
| $\text{SimCLR}_F$ | $\underline{47.9 \pm 0.5}$ | $59.2 \pm 1.5$ | $\mathbf{66.6 \pm 1.5}$ |
| $\text{HFMCA}_F$ | $\mathbf{48.7 \pm 0.8}$ | $59.4 \pm 1.1$ | $\underline{66.0 \pm 1.9}$ |

Table 1: Frozen encoder accuracy (%) on unseen datasets.

| Model | BSNIP | AOMIC (Sex) | HCP (Sex) |
|---|---|---|---|
| Baseline | $47.8 \pm 0.8$ | $62.5 \pm 1.8$ | $\mathbf{71.1 \pm 2.2}$ |
| VICReg | $47.1 \pm 1.0$ | $60.5 \pm 1.7$ | $66.0 \pm 3.2$ |
| BT | $47.2 \pm 1.2$ | $60.3 \pm 1.4$ | $66.1 \pm 1.1$ |
| SimCLR | $\mathbf{48.7 \pm 0.6}$ | $\underline{63.7 \pm 1.8}$ | $68.2 \pm 3.1$ |
| HFMCA | $\underline{48.4 \pm 0.8}$ | $\mathbf{64.6 \pm 1.4}$ | $\underline{70.2 \pm 0.6}$ |

Table 2: Unfrozen encoder accuracy (%) on unseen datasets.

transformations. It decomposes their joint density $\rho(X, Y) = p(X, Y)/(p(X)p(Y))$ into orthogonal components, whose neural approximations $f_\theta(X)$ and $g_\omega(Y)$ learn maximally dependent features. The objective $\mathcal{L}_{FMCA} = \log \det R_{FG} - \log \det R_F - \log \det R_G$ encourages orthogonal features within each view (via $R_F, R_G$) while aligning them across views (via $R_{FG}$).

## 2.2 Problem Setup and Graph Construction

Given a small labelled clinical dataset $D_c = \{(X_i, Y_i)\}$ and a large unlabelled population dataset $D_p = \{X_j\}$, each subject is represented by a functional connectivity matrix $X \in \mathbb{R}^{|V| \times |V|}$ encoding pairwise correlations between brain regions. To mitigate overfitting on $D_c$, we pretrain an encoder $f_\theta$ on $D_p$ with a self-supervised FMCA objective and fine-tune it for diagnosis prediction.

## 2.3 Hierarchical FMCA on Graphs

We adapt FMCA to hierarchical graph representations by contrasting local and global feature dependencies. For a graph $X$ and its augmentations $Y_i = \mathcal{T}_i(X)$, a shared encoder $f_\theta^1$ produces embeddings $Z_i$. These are projected into low-level features $Z_L = [f_\omega^2(Z_1), ..., f_\omega^2(Z_T)]$ and a high-level aggregate $Z_H = \sum_i f_{\omega'}^3(Z_i)$. Low- and high-level features $Z_L$ and $Z_H$ are used to construct correlation matrices $R_L, R_H$ and $R_{LH}$. The objective $\mathcal{L}_{HFMCA} = \log \det R_{LH} - \log \det R_L - \log \det R_H$ maximises statistical dependence between $Z_L$ and $Z_H$, enforcing multi-view consistency across graph augmentations such as random walk sampling, node dropping, and edge perturbation. After pretraining, projection heads are removed, and only the encoder is used for downstream tasks.

We employ a Graph Transformer based on the GPS architecture [16], combining local message passing with global attention. Random Walk Positional Encodings [8] preserve brain region topology, and graph-level embeddings are obtained via global mean pooling.

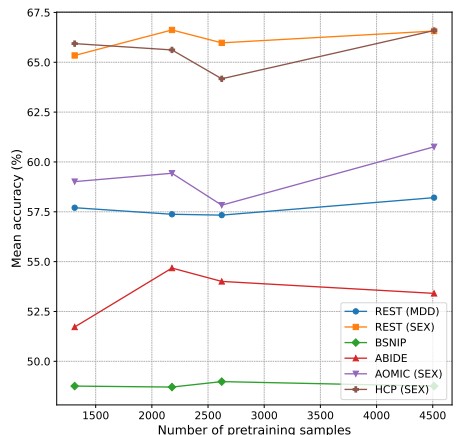

Figure 2: HFMCA pretrained with increasing data volumes. No monotonic scaling trend is observed, consistent with recent findings on negative transfer in graph models.

## 3 Experiments

### 3.1 Experimental Setup

We evaluate our approach on five fMRI datasets covering diverse diagnostic and demographic tasks: **REST** [5]: MDD and sex classification (1642 subjects; 51.6% MDD; 61.0% Male) **ABIDE** [6]: ASD classification (866 subjects; 53.6% ASD) **BSNIP** [19]: Schizophrenia/Bipolar (1464 subjects; 43.7/34.2/22.1% Healthy/SZ/BP) **AOMIC** [18]: Sex (881 subjects; 51.9% Male) **HCP** [9]: Sex (443 subjects; 55.5% Male). Models are pretrained on REST and ABIDE (2000+ subjects) and evaluated on all datasets, including unseen ones (BSNIP, AOMIC, HCP).

The encoder with two projection heads is trained for 200 epochs using Adam ($10^{-3}$ lr, $10^{-6}$ weight decay, batch size 256). After pretraining, projection heads are discarded and the backbone is fine-tuned. We compare HFMCA against SimCLR [4], Barlow Twins [21], VICReg [1], and a randomly initialized baseline. Evaluation follows nested 5-fold cross-validation with 10 runs, using both frozen and unfrozen encoder variants.

### 3.2 Results

Linear classifiers trained on frozen and unfrozen encoders assess transferability. The results (Tables 1 and 2) show that HFMCA consistently outperforms random initialization (*Baseline*) and remains competitive with other methods. Notably, it achieves more stable performance on average, exhibiting lower variance across experimental runs.

To investigate scaling laws, we pretrained HFMCA on increasing dataset sizes: **REST**, **REST+ABIDE**, **REST+ABIDE+HCP**, and **REST+ABIDE+HCP+BSNIP**. As shown in Figure 2, performance peaks for REST+ABIDE and drops when adding more data, consistent with reports of negative transfer in large graph models [12, 2].

## 4 Conclusion

We successfully extended HFMCA to graph-structured fMRI data, providing a theoretically principled approach to self-supervised representation learning. Our method achieves competitive performance across five neuroimaging datasets. The demonstrated transfer learning capabilities and stable training make HFMCA particularly suitable for neuroimaging applications. Future work should explore larger-scale datasets and investigate HFMCA as a component of foundational models for brain imaging. Even though the initial scaling law analysis suggests greater complexity compared to text and vision domains, the framework and transferability indicate important contributions toward generalizable computational models of brain function.

## 5   Potential Negative Societal Impact

This work could enable unintended uses such as inferring cognitive or mental health traits without consent, raising privacy and fairness concerns. Moreover, pretrained models may inherit demographic or dataset biases, leading to inequitable outcomes if applied clinically.

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
