# OpenReview forum: "Adapting HFMCA to Graph Data: Self-Supervised Learning for Generalizable fMRI Representations"
_EurIPS.cc/2025/Workshop/MedEurIPS — EurIPS 2025 Workshop MedEurIPS Submission_

### Official Review · Reviewer_JhHD · 2025-10-31
**Review comments**

**Rating:** 4
**Confidence:** 4

**Review:**

The paper proposes adapting the Hierarchical Functional Maximal Correlation Algorithm (HFMCA) for self-supervised pretraining on graph-structured fMRI data. While the application to neuroimaging is relevant, the technical contribution is limited.

The core of the method is a direct application and adaptation of the existing HFMCA framework to fMRI functional connectivity graphs, which is primarily an engineering effort rather than a methodological breakthrough.

Essential ablation studies is missing. The motivation of choosing a specific garph transformer backbone is not elaborated. It's unclear if the observed competitive performance (Tables 1 & 2) is due to the HFMCA objective, the strong Graph Transformer, or the specific choice of augmentations.

Given the limited technical novelty and the absence of critical ablations to justify the proposed method, the paper does not meet the necessary bar for acceptance at this workshop.

---

### Official Review · Reviewer_Zuq1 · 2025-10-31
**Interesting Adaptation of HFMCA for fMRI Graphs**

**Rating:** 5
**Confidence:** 4

**Review:**

The paper presents an adaptation of HFMCA for self-supervised pretraining on graph-structured fMRI data. While the method largely builds on existing principles, its application to brain connectivity is timely and relevant. The work lacks ablations to isolate where performance gains originate (objective vs. backbone vs. augmentations), and the motivation for the chosen architecture is not fully elaborated. Nonetheless, the direction is promising and could spark valuable discussion on extending dependence-based SSL beyond contrastive formulations.

---

### Decision · Program_Chairs · 2025-10-31

**Decision:**

Reject

**Comment:**

Both reviewers find the adaptation of HFMCA to graph-structured fMRI data relevant and well-motivated, but they agree that the methodological novelty is limited and that key ablations are missing to clarify the source of performance gains.